# Phenological Shifts in a Warming World Affect Physiology and Life History in a Damselfly

**DOI:** 10.3390/insects13070622

**Published:** 2022-07-12

**Authors:** Mateusz Raczyński, Robby Stoks, Frank Johansson, Kamil Bartoń, Szymon Sniegula

**Affiliations:** 1Institute of Nature Conservation, Polish Academy of Sciences, 31-120 Krakow, Poland; kbarton@iop.krakow.pl; 2Department of Biology, Evolutionary Stress Ecology and Ecotoxicology, University of Leuven, 3000 Leuven, Belgium; robby.stoks@kuleuven.be; 3Department of Ecology and Genetics, Animal Ecology, Evolutionary Biology Centre, Uppsala University, 75236 Uppsala, Sweden; frank.johansson@ebc.uu.se

**Keywords:** phenology, life history, immune function, freshwater insect, *Ischnura elegans*, voltinism, climate change

## Abstract

**Simple Summary:**

Climate warming affects phenological events of cold-blooded organisms. In this analysis we studied, in laboratory conditions, the impact of warming and hatching dates on key life history and physiological traits in a cannibalistic damselfly, *Ischnura elegans*. Larvae were reared in groups from hatching to emergence through one or two growth seasons, depending on the voltinism. Larvae were equally divided by hatching dates (early and late) and temperature treatment (current and warming). Early and late hatched groups were not mixed. Despite no difference in cannibalism rate between different hatching dates and temperatures, early hatched larvae reared under warming had elevated immune function measured as phenoloxidase (PO) activity. This increased PO activity was not traded off with life history traits. Instead, age and mass at emergence, and growth rate were mainly affected by temperature and voltinism. Our results confirm the importance of phenological shifts in a warming world for shaping physiology and life history in a freshwater insect.

**Abstract:**

Under climate warming, temperate ectotherms are expected to hatch earlier and grow faster, increase the number of generations per season, i.e., voltinism. Here, we studied, under laboratory conditions, the impact of artificial warming and manipulated hatching dates on life history (voltinism, age and mass at emergence and growth rate) and physiological traits (phenoloxidase (PO) activity at emergence, as an indicator of investment in immune function) and larval survival rate in high-latitude populations of the damselfly *Ischnura elegans*. Larvae were divided into four groups based on crossing two treatments: early versus late hatching dates and warmer versus control rearing temperature. Damselflies were reared in groups over the course of one (univoltine) or two (semivoltine) growth seasons, depending on the voltinism. Warming temperature did not affect survival rate. However, warming increased the number of univoltine larvae compared to semivoltine larvae. There was no effect of hatching phenology on voltinism. Early hatched larvae reared under warming had elevated PO activity, regardless of their voltinism, indicating increased investment in immune function against pathogens. Increased PO activity was not associated with effects on age or mass at emergence or growth rate. Instead, life history traits were mainly affected by temperature and voltinism. Warming decreased development time and increased growth rate in univoltine females, yet decreased growth rate in univoltine males. This indicates a stronger direct impact of warming and voltinism compared to impacts of hatching phenology on life history traits. The results strengthen the evidence that phenological shifts in a warming world may affect physiology and life history in freshwater insects.

## 1. Introduction

Climate change affects organisms’ phenology, i.e., periodic life history events influenced by seasonal and interannual variations in climate [1,2]. Climate change might thus indirectly affect ecological interactions because changes in temperature affect arrival of migrants, egg hatching and other phenological events that may alter competition between organisms [3,4]. Other indirect effects of rising temperatures are increased competition rates, including cannibalism, which can influence population dynamics [5,6]. These changes in organisms’ interactions are expected to happen as species and populations show genetic variation and phenotypic plasticity in phenological events [7,8,9]. At a more mechanistic level, the timing of phenological events and temperature may affect physiology [10,11]. Therefore, variation and plasticity in physiological parameters should be considered when predicting organisms’ responses to global change.

An important type of organismal response to warming is the expression of different developmental trajectories. Alternative developmental trajectories can lead to different ages of sexual maturation. In organisms with complex life cycles such as insects, different developmental trajectories can, in the extreme case, result in cohort splitting [12]. Cohort split occurs when organisms that start their development at the same time follow different, genetically determined physiological pathways that result in different durations of the larval stage. Such cohort splitting was reported in a temperate population of the damselfly, *Calopteryx splendens*, that is, while the predominant fraction of the population completed larval development within one year (univoltine), small fractions took either half a year (bivoltine) or two years (semivoltine) to complete development [13]. In insects, cohort splitting is an important factor in shaping fitness traits such as emergence date, adult body mass and size, and mating success [14]. Cohort splitting might also affect intraspecific competition, especially due to differences in body size, with the slower growing cohort expected to perform worse than the faster growing cohort, and hence reach maturation later [15,16]. In addition, because fast growth leads to earlier maturation and mating, a fast cohort can increase the number of generations per year, and hence fitness components [17,18]. For example, multivoltine (more than two generations per year) populations of the butterfly, *Lycaena hippothoe*, showed higher reproductive success than univoltine populations, and this despite a cost of decreased mass in multivoltine populations [19]. It is, therefore, important to consider cohort splitting when predicting population responses to climate warming because within and between cohort competition for food, space or mating partners can interact with temperature [6,20].

Here, we study how egg hatching phenology and temperature affect juvenile life history, physiology and competition across metamorphosis in high-latitude populations of the damselfly, *Ischnura elegans*. The larvae of damselflies are cannibalistic. The larval cannibalism rate increases when damselflies differ in size [21,22,23], experience prey scarcity [21,24,25] but see [26] and warming temperature [5]. At high latitudes, damselfly populations can have variable voltinism—between one to three years for completing a generation, i.e., uni-, semi- and partivoltine life cycles [27] (Ulf Norling pers. comm.), with the univoltine being more seasonally time constrained than the semi- and partivoltine, i.e., having a shorter time window available (one vs. two or three growth seasons) for development and growth [28,29,30]. We made two predictions. First, we predicted that when reared in a group with early hatchers and in a group with late hatchers, early hatchers will show a higher growth rate, higher mass and lower age at emergence (i.e., increased voltinism), higher investment in immune function measured as phenoloxidase (PO) activity and will experience more competition resulting in decreased survival compared to late hatchers. For early hatchers, early hatching provides more time available for larval development within the growth season and opportunity for completing juvenile development during the next growth season [27], resulting in a fast cohort, i.e., univoltinism [31]. Early hatching should therefore increase development rate and PO activity through increased growth efficiency [7,32,33] and competition over prey [34] compared to late hatchers. This is caused by higher activity and foraging rates of time constrained, univoltine larvae [35]. Late hatchers are predicted to postpone emergence until the following season(s), resulting in a slow cohort, i.e., semi- or partivoltinism, with decreased parameters of development and growth rate, mass at emergence and PO activity [31], and increased survival rate [36]. Second, we predict that warming will increase development rate, growth rate and voltinism, and will decrease mass at emergence and survival rate. Higher temperatures increase metabolism and activity in ectotherms, leading to increased feeding, growth rates, and competition [37,38]. Mass at emergence will be reduced because of a temperature-driven trade-off between age and mass at emergence (temperature–size rule) [39,40]. We further expect that warming will promote increased investment in immune function due to thermal dependence of enzymatic precursor of phenoloxidase (immune function protein), pro-phenoloxidase [41], and will potentially increase pathogen presence at higher temperatures [42,43].

## 2. Materials and Methods

### 2.1. Study Species and Field Collection

*Ischnura elegans* is a common damselfly in Europe. The species occurs from central Sweden to northern Spain [44]. *I. elegans* shows a rapid geographic shift towards the north as a result of climate change [45]. This damselfly has a long aquatic larval stage where most of its growth and development occurs, and thereafter a short terrestrial adult stage with dispersal and reproduction. Larvae hatch 2–3 weeks after egg laying. Larva is the overwintering stage [22]. Individuals with a different voltinism overwinter in various instars [46]. High-latitude populations need 1–3 years for completing larval development, i.e., have a uni- and semivoltine life cycle [27] (Ulf Norling pers. comm.). This is longer than in southern populations which can be bi- or multivoltine, i.e., have two or more generations per year, respectively [47].

To collect eggs, adult *I. elegans* females in tandem with a male were collected at two ponds in central Sweden near Uppsala (59°60′ N 17°40′ E and 59°53′ N 17°38′ E). Females were individually placed in plastic cups with perforated lids and wet filter paper for egg laying at room temperature and the natural photoperiod. Oviposition occurred within three days after females had been field-caught. Eggs were collected after three days, and surviving females were released into their natural populations. Eggs for the early hatching treatment were acquired from females caught 27–30 June 2019, and for the late hatching treatment 13–14 July 2019. In total, 33 early-laid clutches (15 and 18 clutches from the first and second pond, respectively) and 22 late-laid clutches (5 and 17 clutches from the first and second pond, respectively) were collected. Egg clutches were transported to the Institute of Nature Conservation PAS in Krakow, Poland. Clutches from different females and ponds were pooled, creating two groups of eggs, early and late.

### 2.2. Experimental Rearing

The experiment was performed at the Institute of Nature Conservation PAS, Krakow, Poland. *I. elegans* eggs were divided into groups based on their collection dates. Based on date of collection, two hatching phenology groups were created—early (caught in June) and late (caught in July) hatchers. During the first growth season (i.e., the season during in which the eggs were laid), early and late hatched larvae were kept the same temperature (21 °C) and photoperiod (L-D 22–2 h). These conditions reflect natural conditions at the sampling sites at the peak of the growth season, i.e., summer solstice. Temperature treatments were introduced during the following growth season, that is, after the first larval overwintering, marking the start of the experimental treatment. To follow natural temperature changes during the second and third growth seasons, we changed the temperature weekly to follow mean temperature changes in shallow parts of waterbodies (the optimal habitat for damselfly larvae, Corbet 1999) in central Sweden. Temperatures were modelled using the FLake model [8,9,48]. These temperatures are reliable estimates of the field-measured water temperatures in ponds [49]. During the second and third growth seasons, we increased the photoperiod by 2 h, resulting in a L-D 24–0 h (i.e., summer solstice). Based on previous results on temperate damselflies [30], we increased the photoperiod to trigger faster larval growth and development across all treatment groups. Experimental temperatures and photoperiods are shown in Figure 1.

The early and late groups hatched on 16 July and 1 August 2019, respectively. At hatching, clutches of the same date group from the two populations were combined. Next, eight larvae from the same phenology group were placed together in containers (16 cm × 12 cm, height 8 cm) filled with 600 mL of dechlorinated tap water. In these containers the larval density (417 larvae per m^2^) was within the range of larval densities observed for *Ischnura* species in nature [50,51]. Two nylon nets were put inside each container as a substitute for submerged vegetation. On 24 August 2019 we started preparing larvae for winter by gradually decreasing temperature and day length. On 12 September 2019 we started winter conditions of 6 °C and total darkness. We kept larvae in these conditions until initiation of the following spring. On 24 January 2020 we started spring conditions at 12.2 °C (control temperature) and 16.2 °C (warming temperature), and for both thermal treatments the photoperiod was L-D 19:24–4:36 h. The difference of 4 °C matches the predicted mean temperature increase by 2100 under IPCC scenario RCP 8.5 [52]. Temperatures and photoperiods were changed at weekly intervals, thereby keeping the 4 °C temperature difference (Figure 1). A second winter period was induced for larvae by short-term temperature and photoperiod drop at week 55. The short time period of the second winter was used for logistical reasons. One could argue that this may not be sufficient to initiate and end larval diapause, which could affect larval traits measured during the following growth season. However, the emergence distribution after the second wintering shows a peak at week 60 (ca. one month after the second winter period ends, Appendix A). This suggests that the gradual decrease and increase in temperature and photoperiod prior to and after each winter stimulated larvae to enter and then terminate winter diapause. The experiment was concluded when last individual emerged during week 66.

Throughout the experiment, larvae were fed two times per day during the growth seasons and one time every other day during the winter. They received five portions per container of *Artemia salina* nauplii (mean = 201.9 nauplii/portion, SD = 17.2). *A. salina* nauplii are often used as a food source in Odonata research [53,54] due to ease in rearing this food source. Additionally, *I. elegans* are generalists, eating prey appropriately to their occurrence in the field [26]. During the following spring conditions, larvae received eight newly hatched *L. sponsa* as supplementary food. Early and late hatched groups received the supplementary food on 28 January and 10 February 2020, respectively. *L. sponsa* hatchlings originated from eggs acquired from 33 adult females caught at two ponds near Sundsvall, Sweden (62°25′ N 17°16′ E and 62°26′ N 17°21′ E) on 1 August 2019. One day after emergence, adults were weighed and frozen at −80 °C for the physiological analyses.

### 2.3. Response Variables

Larval survival was recorded daily. Individuals that attempted to emerge (dead larvae on the nylon net above the water surface and live, fully emerged adults on nylon nets) were categorized as surviving until emergence. In this study, intrinsic mortality was not verified, and mortality caused by both cannibalism and intrinsic reasons was used to estimate survivorship. However, in a previous group-rearing experiment with *Lestes sponsa*, intrinsic mortality caused by means other than cannibalism was low and did not differ between treatments [55]. Based on this, we assume mortality in *I. elegans* was mainly caused by cannibalism. Development time (i.e., age at emergence) was measured as the number of days between egg hatching and adult emergence, with the exclusion of the winter period(s) where no development is expected [56]. Adult wet mass (mg) was determined one day after emergence by measuring damselfly weight on an electronic balance (Radwag AS.62, Krakow, Poland). Growth rate was calculated by dividing adult wet mass by larval development time. Individuals were classified to one of the two voltinism groups, depending on the emergence season: those that emerged during the second season between week 19 and 46 were considered univoltine, and those that emerged during the third season between week 47 and 66 were considered semivoltine (Figure 1).

Phenoloxidase (PO) activity was quantified from damselfly bodies whose legs and wings had been removed. The bodies were grinded, mixed in phosphate buffer solution (15 µL for each milligram of wet mass) and centrifuged at 10,000× *g* for 5 min at 4 °C. The assay to measure PO activity was based on Stoks et al. (2006). A mixture of 10 µL of homogenate with 105 µL of phosphoric buffered saline (PBS) and 5 µL of chymotrypsin was incubated for 5 min in a 384-well microtiter plate. Afterwards, L-DOPA (1.966 mg dihydroxyphenyl-L-alanine per 1 mL of PBS-buffer) was added to the samples, followed by immediate measurement of the linear increase in absorbance at 490 nm every 20 s for 30 min at 30 °C. The average values of the slope of the linear part of the reaction curve from two technical replicates was used for the statistics. To correct the PO activity, the protein content of the samples was measured using a modified Bradford [57] procedure. A mixture of 1 µL of homogenate, 160 µL of Milli-Q-water and 40 µL of Bio-Rad Protein Dye was incubated for 5 min at 25 °C. Afterwards, the absorbance at 595 nm was measured and converted into protein content using a standard curve of bovine serum albumin. The averages of three technical replicates per larva were used for statistical analyses. To express the activity of PO per total protein content, the values of slope of the reaction curve for PO was divided by the values of protein content.

### 2.4. Statistical Analyses

To analyze the response variables of survival until emergence and voltinism, generalized mixed models with a binomial error distribution were used. In these models early versus late hatching, control versus warming, and sex were fixed effects. Note that voltinism was not included as an explanatory variable in the survival analysis because survival was estimated at emergence only, and not between growth seasons. Other response variables (age and mass at emergence, growth rate and PO activity) were analyzed using linear mixed models. In these models hatching phenology, temperature, sex and voltinism were included as fixed effects. Container was included as a random effect to account for multiple larvae sharing the same container. At first, models with all possible interactions were performed, then all interactions with *p* ≥ 0.05 were removed at once. Final models included all fixed effects and interactions with *p* ≤ 0.05. Post-hoc Tukey HSD tests were used to assess pairwise between-level differences. To perform statistical analyses, we used R 4.0.4 software (Krakow, Poland), with packages lme4 [58] for mixed-effect modelling and emmeans [59] for post-hoc tests.

## 3. Results

In total, 113 (12% per total at the start) larvae attempted to emerge, and among these 37 (32.7%) individuals were univoltine and 76 (67.3%) individuals were semivoltine. Out of 92 larvae that emerged with success, 35 (38.0%) were univoltine and 57 (62.0%) were semivoltine (Appendix A). The percentage of emergence attempts that were successful per total attempts was 81.4%. More univoltine (94.6%) than semivoltine (75.0%) individuals attempting to emerge did so successfully (χ^2^ ≤ 5.036, *p* ≥ 0.025). Hatching phenology, temperature and sex did not affect survivorship (Figure 2A, Table 1). Early versus late hatching did not affect voltinism (Figure 2B, Table 1). Under warming more univoltine than semivoltine individuals emerged, with the opposite happening at the control temperature (post-hoc contrasts for semivoltine/univoltine ratio, early versus late hatching date and control versus warming temperature: early hatching, control temperature—early hatching, warming temperature *p* < 0.01; late hatching, control temperature—late hatching, warming temperature *p* < 0.01; early hatching, warming temperature, late hatching, control temperature *p* < 0.01, Figure 2B, Table 1). Males showed equal ratio of uni- and semivoltine individuals, whereas females tended to have higher ratio of semivoltine individuals (Appendix A, Table 1). Body mass was not affected by hatching phenology (Figure 2C, Table 1). Temperature and voltinism affected mass in an interactive way: univoltine individuals were heavier than semivoltine individuals, but only at the control temperature. Warming decreased mass, but only in univoltine individuals (interaction temperature × voltinism, Figure 2C and Appendix A, Table 1). Females were heavier than males (Appendix A, Table 1). Early hatched individuals took longer to develop than late hatched ones (Figure 2D and Appendix A, Table 1). Hatching phenology did not affect development and growth rate (Figure 2D,E, Table 1). Temperature, sex and voltinism affected both development time and growth rate in an interactive way. Warming shortened development time, but only in univoltine females (interaction temperature × sex × voltinism, Appendix A, Table 1). Warming in univoltine females increased growth rate, while in univoltine males it decreased growth rate (interaction temperature × sex × voltinism, Appendix A, Table 1).

Hatching phenology and temperature affected PO activity in an interactive way. Early hatchers showed increased PO activity, but only in the warming treatment (interaction hatching phenology × temperature, Figure 2F and Appendix A, Table 1). Females had a higher PO activity than males, but only in the warming treatment (interaction temperature × sex, Appendix A, Table 1). PO activity was not affected by voltinism (Figure 2F, Table 1).

## 4. Discussion

Here we explored how hatching date and temperature affected key life history traits such as survival and age and mass at emergence, and immune function at emergence in a damselfly. Given the predicted climate change scenarios in the future, it is important to understand how shifts in life history events and environmental conditions affect these individual traits. Our results showed a complex interplay between phenology and warming. This suggests that it may not be easy to predict how future climate change will affect population abundance and dynamics.

We predicted that early hatched individuals should have a higher mass at emergence compared to late hatching individuals. We found no support for this prediction, but we found that voltinism and temperature affected this trait. Univoltine damselflies showed a higher mass than semivoltine individuals, but this difference disappeared under warming. This result is in contrast with the compound interest hypothesis, which states that warming increases the number of generations per season, but with a cost of a lower mass at maturation [19,60,61]. On the other hand, adding an extra season for completing a generation (i.e., here, a switch from uni- to semivoltine life cycle) should lead to a greater mass and/or size at emergence [28], hence increased reproductive success [62,63] but see [64,65]. In our case, the opposite was found. This suggests a two-fold potential fitness benefit for univoltine ischnurids: a shorter generation time and a larger mass at emergence. The effect of mass on fitness might be expressed by greater emergence success per emergence attempt in univoltines compared to semivoltines. A previous study on ischnurids indicated that emergence success was positively correlated with mass in the final larval instar [66], which in turn correlates with other fitness traits in adult insects, e.g., structural size and fecundity [22,67,68].

Interestingly, hatching phenology did not affect voltinism, but warming increased the ratio of univoltine individuals, confirming that warming will increase the number of generations per year in ectotherms, also those living at high latitudes. However, at some point this trend for increased voltinism might become less beneficial, and this is because of a decreased mass in fast developing damselflies [69], resulting in a decreased fitness.

The temperature-dependent difference in mass in univoltine ischnurids supports the temperature–size rule (TSR), where warming leads to acceleration of development rate without relative increase in growth rate, leading to smaller mass and size at maturation [39,70]. Our results on development and growth rates add support for this rule. The above suggests that warming will reduce mass at emergence through increased development rates, as is generally expected in ectothermic organisms [70,71], and that advanced hatching dates will have minor effects on age and mass at emergence. Interestingly, the TSR pattern did not occur in semivoltine individuals. This result supports the compound interest hypothesis, where organisms with decreased voltinism are expected not to react or to react weakly to increasing temperature by changing their development rates and mass at emergence (for example, a weak TSR response in univoltine population of a butterfly *Lycaena hippothoe* compared to multivoltine population [19]).

We found a trend for a positive relationship between age and mass at emergence within the univoltine cohort: the lighter individuals emerged at the beginning of the growth season (a trend for a positive correlation between age and mass at emergence, r = 0.34, *p* = 0.092, Appendix A). Such positive correlation is rather rare in insects, including odonates [72,73], and the opposite pattern is commonly reported [74,75,76]. Within and across seasons, temporal variation in mass at emergence is not well understood, and both internal (physiological) and external (ecological, e.g., competition and temperature) factors were suggested to explain the pattern [28,77,78]. In our case, a likely explanation, though not quantified in this study, might be that larvae from the univoltine cohort showed higher intraspecific competition for *Artemia* nauplii (mild form of competition), but not resulting in cannibalism (extreme form of competition) [78].

We did not find support for our prediction of a lower survivorship in the early hatching group. Similar rates of survivorship between both phenology groups could be explained by the absence of strong competition, including cannibalism between different size cohorts. Size-dependent cannibalism caused by, for example, differences in hatching dates within rearing group, was reported in previous studies [23,55,79,80]. Our results support the theoretical prediction indicating that the outcomes of interactions between individuals from different cohorts with different trait values depends less on the hatching dates and more on other population characteristics, e.g., population density [81]. In our study both cohorts were present within each phenology group. Differences in development rates between overlapping cohorts might have reduced initial differences in larval sizes caused by different hatching dates between early and late hatched groups, which covered about 8% of the growth season [48]. Overlapping cohorts within each of the hatching phenology group could weaken hatching phenology effects on life history traits, as it likely happened in the current study.

In ectotherms, an increased temperature may imply an increased immune challenge because of increased activity and growth of pathogens [43,82,83,84]. However, in our study no specific immune challenge was posed. In *I. elegans*, increasing temperatures up to 30 °C can increase the investment in immune function [85]. Such increase in PO activity was observed in early hatched females reared under warming. The increased PO activity at the higher temperatures might be caused by relocation of resources from other physiological traits such as energy reserves [42,86], but this needs further study for confirmation. However, this positive effect of warming on immune function might be paired with a response to thermal stress resulting in, for example, decreased survival in ectotherms [87,88]. Our results suggest that advanced hatching due to warming might be beneficial for the damselfly, given the increased energy investment into immune function is not traded off with investments in other physiological functions.

We found the highest PO activity in early hatched females grown under experimental warming. Such response may be beneficial for females because an elevated immune function can lead to increased longevity, the major determinant of female reproductive success [89,90]. On the other hand, because investment in immune function is costly, males are expected to invest more in traits which lead to increased mating success, e.g., increased activity in searching for a mate, and investment in secondary sexual traits such as ornamentation [89,90,91]. Hence, advanced hatching due to warming would promote increased immune reaction to pathogens in both sexes, and more so in females than males.

In conclusion, our results demonstrate that under experimental warming early hatched *I. elegans* showed an increased investment in immune function that, in companion with increased development and growth rate, may indicate an adaptive response to climate change in the high-latitude, season-limited populations. Our findings highlight the importance of variable voltinism associated with cohort splitting, even among individuals with the same hatching period, and how this might affect key fitness traits of ectotherms in a warming world.

## Figures and Tables

**Figure 1 insects-13-00622-f001:**
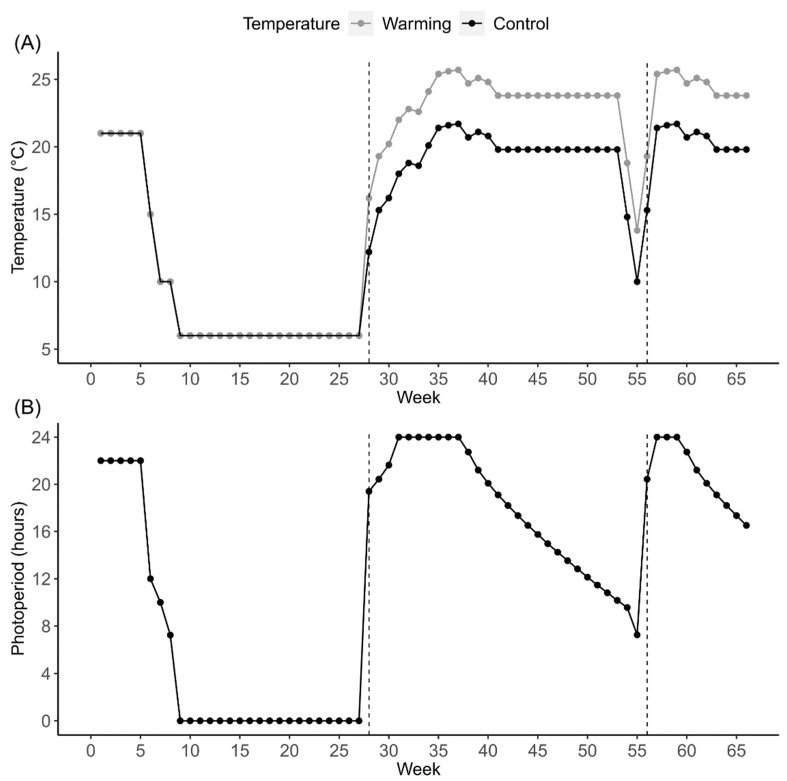
(**A**) Temperatures in °C and (**B**) photoperiods in hours of light during a 24 h cycle used during the experiment. Temperature treatment groups differed by 4 °C. The photoperiod was kept identical in both temperature treatments. The first growth season lasted from week 1 to week 8. The second growth season lasted from week 28 to week 55 (start is indicated by a dashed line to the left). The third growth season lasted from week 56 to week 66 (start is indicated by a dashed line to the right). During weeks 9 to 27 we installed winter conditions at 6 °C and total darkness. The short-term temperature and photoperiod drop at week 55 simulated the second winter condition.

**Figure 2 insects-13-00622-f002:**
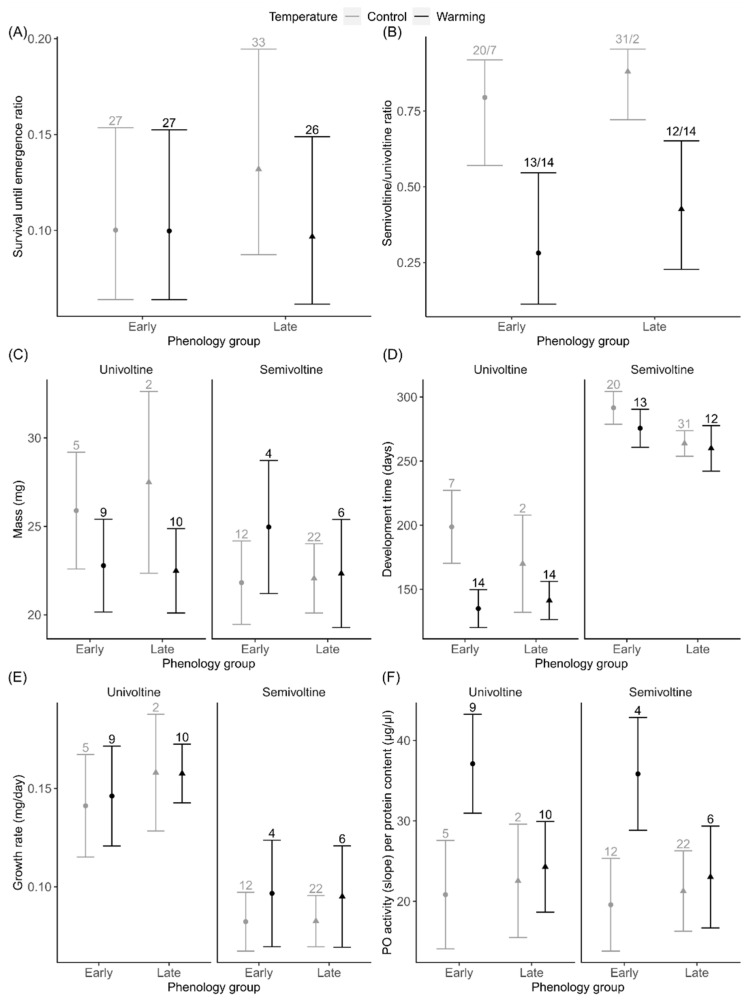
The effects of hatching phenology and warming on the life history and physiology of univoltine and semivoltine *Ischnura elegans*: (**A**) larval survival until emergence (ratio of surviving to all); (**B**) voltinism (ratio of semivoltine to univoltine); (**C**) mass at emergence; (**D**) development time; (**E**) growth rate; (**F**) phenoloxidase (PO) activity. Note that the voltinism effect was excluded from the survival analysis because survival was estimated at emergence only, and not between growth seasons. Error bars indicate estimated 95% CI. The numbers at the top of the error bars represent the number of damselflies within each group.

**Table 1 insects-13-00622-t001:** Results from mixed models testing for effects of hatching phenology, temperature, sex and voltinism on *Ischnura elegans* survival, voltinism, mass, development time, growth rate and phenoloxidase (PO) activity across emerged adults. Voltinism was analyzed as a response variable and as one of the explanatory factors when analyzing other traits. Note that the voltinism effect was excluded from the survival analysis because survival was estimated at the emergence only, and not between growth seasons. Final models included all fixed effects and interaction terms with *p*-values ≤ 0.05. *p*-values below 0.05 are in bold.

Predictor	Df	χ^2^	*p*
Survival			
Hatching phenology	1	0.379	0.548
Temperature	1	0.583	0.444
Sex	1	0.094	0.760
Voltinism			
Hatching phenology	1	1.078	0.299
Temperature	1	14.235	**<0.001**
Sex	1	3.155	0.076
Mass			
Hatching phenology	1	0.616	0.433
Temperature	1	0.468	0.494
Sex	1	13.651	**<0.001**
Voltinism	1	7.516	**0.006**
Temperature × voltinism	1	7.216	**0.007**
Development time			
Hatching phenology	1	11.437	**<0.001**
Temperature	1	7.526	**0.006**
Sex	1	0.280	0.600
Voltinism	1	385.402	**<0.001**
Temperature × voltinism	1	4.345	**0.037**
Temperature × sex	1	3.720	0.054
Temperature × sex × voltinism	2	6.293	**0.043**
Growth rate			
Hatching phenology	1	0.095	0.758
Temperature	1	0.003	0.954
Sex	1	4.455	**0.035**
Voltinism	1	158.661	**<0.001**
Temperature × sex	1	3.729	0.053
Temperature × sex × voltinism	3	13.344	**0.004**
PO activity			
Hatching phenology	1	4.999	**0.034**
Temperature	1	10.178	**0.001**
Sex	1	2.639	0.105
Voltinism	1	1.630	0.201
Hatching phenology × temperature	1	15.047	**<0.001**
Temperature × sex	1	5.963	**0.015**

## Data Availability

The data presented in this study are available in Appendix A (Impact of hatching phenology on immune function, and voltinism and temperature on life history.xlsx).

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
