# Peer review of "Phenological Shifts in a Warming World Affect Physiology and Life History in a Damselfly"

_insects, 2022, doi:10.3390/insects13070622_

Round 1

Reviewer 1 Report

This manuscript describes results from a lab experiment whose goals were to determine how higher ambient temperature and hatching date affect larval survival and growth rates and therefore age and mass at emergence, and phenoloxidase activity of emerged adults of Swedish population of Ischnura elegans. Relative to the control treatment (i.e. ‘current temperature’), an increase of 4 degrees C. during larval development did not affect larval survivorship but did result in more adults emerging after one season as opposed to two seasons as currently occurs for the Swedish population. A hatching difference of 2 weeks had no effect on the number of generations/season (voltinism) but  larvae hatching 2 weeks earlier had higher PO activity as adults, independent of age or mass at emergence. The limited nature of the results do not justify the conclusion that they ‘confirm the importance of phenological shifts in a warming world for shaping physiology and life history in a freshwater insect.’  They are consistent with those predictions. No rationale for the predictions to be tested were provided by the authors.

Lab experiments can control specific variables whereas the comparative method applied to natural populations cannot, but in this case it seems that citing comparative studies would at least provide the current missing rationale for the predictions that are being tested. We already know from comparing the semivoltine populations of I. elegans in Sweden to the multivotine populations found in southern Europe  (e.g. work by Cordero et al. and others), that with continued global warming, northern populations will become multi-voltine. Similarly one can point to latitudinal comparisons of differences in adult size of I. elegans (and perhaps even in PO activity ?) to predict what would happen under global warming.

The title is confusing and needs to be more specific – e.g. Stong impact of temperature on voltinism, etc. It is distracting to mention a ‘cannibalistic damselfly’ when all odonates are cannibalistic given the right circumstances.  No evidence for cannibalism was ever presented in the results; survivorship was measured, as is correctly indicated in Table 1. No information on natural diet or density were provided (i.e.unless I’m mistaken,  Corbet 1999 doesn’t seem to provide anything specific with respect to I. elegans density or diets). What study shows that the density used in this experiment  (417 larvae/ m2) was ‘within the range of larval densities observed’? In contrast, Banks and Thompson (1987) report wild densities of  I. elegans nearly 3 times as great as those used in the experiment. Moreover, several authors cited in Corbet (1999) incate significant mortality in the early developmental stages of individually reared odonates (i.e. with no ability to cannibalize).

The abstract is overblown;  the implication that ‘finding a significant effect of warming on these traits should affect ‘ecological interactions’ is pure speculation. The authors present no evidence of any ecological interactions. If one wants to address how climate change is ‘altering ecological interactions’ comparative field studies are desperately needed. Indeed, this natural experiment is progressing faster than climate scientists predicted.

The manuscript seems to have been submitted prematurely; e.g. authorship affiliation is lacking for 3, the bibliography is inconsistent in format, parts of the text, including the title, are confusing (see below).

Below are more specific questions along with suggested changes to improve clarity of the ms. Numbers refer to lines of text.

13-14 change to ‘We studied, under laboratory conditions, the impact…..’

17 change to : Larvae were divided into four groups of two treatments each:  early versus later hatching dates and warmer versus the current temperature.

18 if you make the above change, you can eliminate ‘Early and late hatched groups were not mixed’.

19 change to ‘cannibalism’ to ‘survival’   and change ‘,’ to ‘.’ And ‘early’ to ‘Early’…..

21 change to read ‘….activity did not affect growth rate and mass at emergence. The latter were… here please tell the reader HOW they were affected by temperature and the number of generations per season.

22 change ‘confirm’ to ‘are consistent with’….

25 change ‘advance their phenological events’ to ‘hatch and grow faster…..’

38 change ‘canniablism rate’ to ‘survivorship’, or age and mass at emergence. ……

deleleting  ‘was not traded off with life history traits’ in preference to the specific results.

39 be specific, i.e. Age and mass at emergence decreased with warming while growth rate increased.

41 change to ‘Our results strengthen the conclusion that a warming world should affect physiology and life history traits ….etc. ‘  or something similar

51 change ‘will’ to ‘may’….l.

52 please add a ref after ‘…organism’

52 delete ‘d’ to read ‘increase competition….’

57 delete ‘also’ unless you specifiy in addition to what?

60 replace ‘dates’ with the more general term ‘younger ages’ of sexual maturation and delete breeding time as it goes without saying that if you decrease the age of maturity organisms willl breed earlier (no actual breeding was done).

63 please explain what is meant by cohort splitting here, since its used to explain several things and not all readers may understand the term.

70 change to read  ‘beneftied in terms of ‘ to ‘increased’ reproductive success by ‘decreasing generation time’.

71 – 73 please be more clear here in telling the reader specifically what you mean – i.e. specifically, what kinds of ‘interactons’?

76. you need a reference telling the reader just how ‘cannibalistic’ they are. With a few exceptions of odonates in small microhabitats, many dragonflies cannibalize only when food is limiting.

78 – Explain the rationale behind each prediction. Otherwise, it sounds like you are just predicting the results you found.

79 what is a ‘phenology uniform group’? early or later hatchers with a given voltinism?

82-84 Please explain what you mean here.

89 change ‘cannibalism’ to ‘survivorship’ because that is what was actually measured.

97-98 I think this might be true for the Swedish study population, but certainly not in more southern populations with multiple generations per season. What is the lifespan of adults in Sweden?

106 replace sentence with ‘Eggs were collected after three days, and females released’. So no females died? I assume larvae were pooled across females so female ID was not a random effect, correct? You might mention that in the methods.

FIG 1.  Seems better to describe the ‘treatments’ as ‘Warmer’ versus ‘Ambient Control’ or just ‘Control’ and explain in your methods that ‘controls’ are those reared at current ambient temperatures. ‘Current’ seems a bit odd.

143-145 this is confusing. Were 8 larvae each placed in its own ‘individual container’? this specifies that there was only 1 larva per container.

146 please replace the Corbet ref. (which is not helpful as it doesn’t specify data related to Ischnura or even coenagrionids) with one that actually shows what natural densities of I. elegans are in Sweden or elsewhere.  And the entire idea of assuming that any death was due to cannibalism, if true, could also be a function of the feeding regime. Do you have any idea of the natural rate of prey capture is for your population?

156-159  All of this is to justify the very brief drop in temperature shown in Fig. 1. I think you at least need to mention to the reader why that ‘winter’ depicted in Fig. 1 is so short.

160-165 How do you justify the diet used? See comment 146 above.

167 change weighted to ‘weighed’

170 were all those ‘dead’ larvae final instars? That seems to be an important point for your conclusions.

172 Correction needed here – wasn’t the paper cited (line 174) about  Lestes sponsa, a damselfly with a different life history from I. elegans?  

183 change to read ‘….damselflies whose legs and …’

203 for clarity, change  ‘hatching phenology’ and temperature to read  ‘early versus late hatching, warming versus control and sex were fixed effects’…..This makes the dichotomy in each treatment clear.

203 remind the reader here what the response variables were – i.e. survivorship, age and mass at emergence, etc.

205 delete ‘the’ to read ‘….at emergence only…’

206 add ‘models’ after ‘using linear mixed’.

208 change to read ‘…for multiple larvae sharing….’

217-218 confusing sentence

223 change ‘survival’ to ‘surviorship’ and end the sentence there; the rest is repetitive as cannibaslim was simply assumed.

223 replace hatching phenology with ‘Early versus late hatching ….

228 delete ‘n’ to read ‘higher ratio’.

228 Fig. 2B doesn’t show anything by sex

Fig 2. In B can you show which contrasts were significant ? (maybe with an asterix)

264 Please be more specific than ‘…life history and physiology’.

274 change to read ‘in contrast with …’

277-280 I would have thought adding a second generation to a season would result in lower mass and size at emergence because growth is speeded up. So at some point with global warming, even in Sweden it might no longer be beneficial.

303. change to read ‘is rather rare in insects…’

311  well, the difference between early and later hatching was only a 2-week difference. Perhaps tell us what % of the total growing season that is in Sweden, which would make it sound more significantly different.

Author Response

This manuscript describes results from a lab experiment whose goals were to determine how higher ambient temperature and hatching date affect larval survival and growth rates and therefore age and mass at emergence, and phenoloxidase activity of emerged adults of Swedish population of Ischnura elegans. Relative to the control treatment (i.e. ‘current temperature’), an increase of 4 degrees C. during larval development did not affect larval survivorship but did result in more adults emerging after one season as opposed to two seasons as currently occurs for the Swedish population. A hatching difference of 2 weeks had no effect on the number of generations/season (voltinism) but  larvae hatching 2 weeks earlier had higher PO activity as adults, independent of age or mass at emergence. The limited nature of the results do not justify the conclusion that they ‘confirm the importance of phenological shifts in a warming world for shaping physiology and life history in a freshwater insect.’  They are consistent with those predictions. No rationale for the predictions to be tested were provided by the authors.

>>>We agree that our experimental setup makes our results only consistent with these predictions, and do not confirm our previous conclusions. We included additional reasoning for our predictions (L: 86-115) and changed our final conclusions (L:38-40, L:414-417) so they are more appropriate with our experimental setup and results (for example , “The results strengthen the evidence…”, L:38).

Lab experiments can control specific variables whereas the comparative method applied to natural populations cannot, but in this case it seems that citing comparative studies would at least provide the current missing rationale for the predictions that are being tested. We already know from comparing the semivoltine populations of I. elegans in Sweden to the multivotine populations found in southern Europe  (e.g. work by Cordero et al. and others), that with continued global warming, northern populations will become multi-voltine. Similarly one can point to latitudinal comparisons of differences in adult size of I. elegans (and perhaps even in PO activity ?) to predict what would happen under global warming.

>>>We provided additional references to support and extend our reasoning, including effects of temperature on adult size of I. elegans, L:86-115. In general our hypothesis has been majorly restructured. Because of early hatching date, and hence more time available for larval development within the growth season, early group should complete larval development within one year, i.e., univoltine life cycle (L:97-103). However, in general these early hatched univoltine individuals are expected to be more seasonally time constrained than late hatched semi- or partivoltine individuals, and this because the earlier have overall fewer time to complete larval development (L:87-91). This should also cause increased activity and foraging rates (L:102-103), leading to increased mass and phenoloxidase activity investment. We also followed a general predictions of what would happen under global warming, specifically that trade-off will occur between increased development rate and decreased mass, L:110-111. PO activity should increase because phenoloxidase is temperature-dependant, and warmer temperature promote increased presence of pathogens (L:111-115).

The title is confusing and needs to be more specific – e.g. Strong impact of temperature on voltinism, etc. It is distracting to mention a ‘cannibalistic damselfly’ when all odonates are cannibalistic given the right circumstances.  No evidence for cannibalism was ever presented in the results; survivorship was measured, as is correctly indicated in Table 1. No information on natural diet or density were provided (i.e.unless I’m mistaken,  Corbet 1999 doesn’t seem to provide anything specific with respect to I. elegans density or diets). What study shows that the density used in this experiment  (417 larvae/ m2) was ‘within the range of larval densities observed’? In contrast, Banks and Thompson (1987) report wild densities of  I. elegans nearly 3 times as great as those used in the experiment. Moreover, several authors cited in Corbet (1999) indicate significant mortality in the early developmental stages of individually reared odonates (i.e. with no ability to cannibalize).

>>>Thank you for this important comment. We agree that in case of I. elegans initial larval density was not high, however according to the literature, larval density was within the range of densities reported in nature. We added appropriate references, L:173-175.We changed the description of mortality. We did not exclude the possibility of cannibalism, but we also mention intrinsic mortality as possible cause (L:208-216). We provided additional information about diet, that died used by us is often used in odonate research. We also mention that I. elegans has a generalist diet (L:196-199). We revised the title considerable, e.g. by removing cannibalism (new title: Phenological shifts in a warming world affects physiology and life history in a damselfly).

The abstract is overblown;  the implication that ‘finding a significant effect of warming on these traits should affect ‘ecological interactions’ is pure speculation. The authors present no evidence of any ecological interactions. If one wants to address how climate change is ‘altering ecological interactions’ comparative field studies are desperately needed. Indeed, this natural experiment is progressing faster than climate scientists predicted.

>>>We agree on this comment. We did not directly measure behavioural and other interaction traits. However, we provided additional context and reasoning for our hypothesis, which has been mentioned in our previous comment, L: 86-115. We also changed abstract, including terms’ descriptions (L:22-25, L:31-35). We removed the references to “ecological interactions”, we did not measure these interactions (L:18). However, we decided to keep this term in the introduction (L:48, L:52) as a background for our study. We also made the reasoning less speculative, e.g., by changing the concluding line in the abstract to be more in agreement with our findings (“The results  strengthen the evidence…”, L:38-40) and attributing survival not only to cannibalism, but also intrinsic mortality (L:28-216).

The manuscript seems to have been submitted prematurely; e.g. authorship affiliation is lacking for 3, the bibliography is inconsistent in format, parts of the text, including the title, are confusing (see below).

>>>We have provided additional information for authorship affiliation and verified consistency in bibliography. We also changed the title (new title: Phenological shifts in a warming world affects physiology and life history in a damselfly)

Below are more specific questions along with suggested changes to improve clarity of the ms. Numbers refer to lines of text.

13-14 change to ‘We studied, under laboratory conditions, the impact…..’

>>>We changed this sentence in accordance to the suggestion, L:18-19.

17 change to : Larvae were divided into four groups of two treatments each:  early versus later hatching dates and warmer versus the current temperature.

>>>We changed this sentence in accordance to the suggestion (“Larvae were divided into four groups based on crossing two treatments: early versus late hatching dates and warmer versus control rearing temperature”), L:22-25.

18 if you make the above change, you can eliminate ‘Early and late hatched groups were not mixed’.

>>>We have made changes according to the suggestions, resulting in phrasing citied in previous comment, L:22-25.

19 change to ‘cannibalism’ to ‘survival’   and change ‘,’ to ‘.’ And ‘early’ to ‘Early’…..

>>>We followed this suggestion and made changes, i.e. changing cannibalism to survival rate (L:22). We also changed “cannibalism” to “survival” or “survivorship” unless we specifically mean cannibalism, L:28.

21 change to read ‘….activity did not affect growth rate and mass at emergence. The latter were… here please tell the reader HOW they were affected by temperature and the number of generations per season.

>>>We changed this sentence to “Increased PO activity was not associated with effects on age and mass at emergence and growth rate.”, L:31-33. We have provided description of how these variables affected aforementioned traits (“Warming decreased development time and increased growth rate in univoltine females, yet decreased growth rate in univoltine males.”), L:33-36.

22 change ‘confirm’ to ‘are consistent with’….

>>>We included this change, making this sentence less speculative (The results strengthen the evidence…”), L:38-40.

25 change ‘advance their phenological events’ to ‘hatch and grow faster…..’

>>>We implemented this change in the abstract, L:16-17.

38 change ‘canniablism rate’ to ‘survivorship’, or age and mass at emergence. ……

>>>We changed the term “cannibalism rate” to “survival rate”, L:22.

deleleting  ‘was not traded off with life history traits’ in preference to the specific results.

>>>We have deleted this sentence as suggested, and added “Increased PO activity was not associated with effects on age and mass at emergence and growth rate…”, L:31-36.

39 be specific, i.e. Age and mass at emergence decreased with warming while growth rate increased.

>>>We included the specific results in the description according to this and previous comment, (“Warming decreased development time and increased growth rate in univoltine females, yet decreased growth rate in univoltine males.”), L:33-36.

41 change to ‘Our results strengthen the conclusion that a warming world should affect physiology and life history traits ….etc. ‘  or something similar

>>>According to this and previous suggested we have changed this sentence to make it in agreement with our results, (“The results  strengthen the evidence…”, L:38-40).

51 change ‘will’ to ‘may’….l.

>>>We included this change as suggested, L:49.

52 please add a ref after ‘…organism’

>>>We added references according to the topic of this sentence, L:50.

52 delete ‘d’ to read ‘increase competition….’

>>>We made the suggested change, L:50

57 delete ‘also’ unless you specifiy in addition to what?

>>>We put “also” here in order to specify that physiological parameters are important in addition to phenology and life history. Since it should be clear from the context we followed the suggestion, L:56.

60 replace ‘dates’ with the more general term ‘younger ages’ of sexual maturation and delete breeding time as it goes without saying that if you decrease the age of maturity organisms willl breed earlier (no actual breeding was done).

>>>We followed this suggestion and changed the sentence to “different ages”, L:59-60.

63 please explain what is meant by cohort splitting here, since its used to explain several things and not all readers may understand the term.

>>>We provided an explanation concerning the term cohort splitting (“Cohort split occurs when organisms which start their development at the same time follow different, genetically determined physiological pathways that result in different durations of the larval stage”) and provided an example of cohort splitting based on Calorypteryx splendes, L:62-68.

70 change to read  ‘beneftied in terms of ‘ to ‘increased’ reproductive success by ‘decreasing generation time’.

>>>We changed this sentence to “showed higher reproductive success than univoltine…” and generally restructured this part to avoid confusion between two groups (instead of increased/decreased voltinism we used multivoltine or univoltine), L:74-78.

71 – 73 please be more clear here in telling the reader specifically what you mean – i.e. specifically, what kinds of ‘interactons’?

>>> We added examples of interactions (“within- and between cohort competition for food, space or mating partners”) to specify the exact kind of competition, L:78-81.

  1. you need a reference telling the reader just how ‘cannibalistic’ they are. With a few exceptions of odonates in small microhabitats, many dragonflies cannibalize only when food is limiting.

>>>We have added more information and references about cannibalism, listing factors which promote cannibalism, i.e. prey scarcity, size difference and warming, L:84-86.

78 – Explain the rationale behind each prediction. Otherwise, it sounds like you are just predicting the results you found.

>>>We provided additional explanation under each of our predictions, L: 86-115.

79 what is a ‘phenology uniform group’? early or later hatchers with a given voltinism?

>>>We changed this term to “in a group within the early hatchers and in a group within the late hatchers” so it is clear what it is meant by this term, L:92-93.

82-84 Please explain what you mean here.

>>>We meant that early hatched larvae have more time for growth within the growth season. This can result in finishing their development and emergence during the following growth season, i.e. univoltine cohort, L:97-103.

89 change ‘cannibalism’ to ‘survivorship’ because that is what was actually measured.

>>>We have changed this term to survival rate, L:108.

97-98 I think this might be true for the Swedish study population, but certainly not in more southern populations with multiple generations per season. What is the lifespan of adults in Sweden?

>>>We added information about voltinism in low latitude populations. However, this experiment restricted to high latitude populations, so the comparison of voltinism between latitudes is not relevant to our study, L:125-129.

106 replace sentence with ‘Eggs were collected after three days, and females released’. So no females died? I assume larvae were pooled across females so female ID was not a random effect, correct? You might mention that in the methods.

>>>We changed the sentence according to suggestion, as well as provided information that surviving females were released to the wild, L:134-136, 141-142.

FIG 1.  Seems better to describe the ‘treatments’ as ‘Warmer’ versus ‘Ambient Control’ or just ‘Control’ and explain in your methods that ‘controls’ are those reared at current ambient temperatures. ‘Current’ seems a bit odd.

>>>We changed the description of the treatment from current to control in the manuscript and supplementary material, i.e. L:24, L:180, L:245

143-145 this is confusing. Were 8 larvae each placed in its own ‘individual container’? this specifies that there was only 1 larva per container.

>>>We have corrected the description to simply “were places together in containers” to indicate that larvae were kept in groups of 8 larvae per container, L:172.

146 please replace the Corbet ref. (which is not helpful as it doesn’t specify data related to Ischnura or even coenagrionids) with one that actually shows what natural densities of I. elegans are in Sweden or elsewhere.  And the entire idea of assuming that any death was due to cannibalism, if true, could also be a function of the feeding regime. Do you have any idea of the natural rate of prey capture is for your population?

>>>We have changed the source of our information about larval densities, L:175. Based on larval densities in nature described in Banks and Thompson, Fig. 5, (1987), I. elegans density varied, depending on the sampling date, and ranged between ~200 and ~600 per m2, which is close to our densities at the start of the experiment. Similarly larval densities from nature were reported by Uttley (1980). This shows that we used densities which are representative of what occurs in nature.

While we do not have data for natural rate of prey capture for our populations, we have provided ad libitum food during the whole experiment. This was indicated by leftover of Artemia nauplii in the containers. Artemia nauplii  are often used as food in experiments on odonates. We included this information, L:196-199.

156-159  All of this is to justify the very brief drop in temperature shown in Fig. 1. I think you at least need to mention to the reader why that ‘winter’ depicted in Fig. 1 is so short.

>>>We added explanation that the duration of second winter was short mainly due to logistical reasons. While the short winter period may have affected diapause and larval traits, this is unlikely to have occurred in our experiment due to a peak of emergences at week 60, ca. one month after the end of the second winter. This suggests that gradual changes of temperature and photoperiod before and after winter lead to starting and terminating winter diapause. L:184-192.

We made a mistake in the original version of Fig. 1 (omission of 9 weeks during the first winter). This mistake affected the description of the figure and methods. We apologise for this mistake and we corrected it accordingly.

160-165 How do you justify the diet used? See comment 146 above.

>>> In general odonates have a broad, generalist diet. However, during this kind of experiments Artemia nauplii are often used. We based our experiment on previously established feeding method. We added this information, L:196-199.

167 change weighted to ‘weighed’

>>>We have included this change as suggested, L:204.

170 were all those ‘dead’ larvae final instars? That seems to be an important point for your conclusions.

>>>We clarified that we counted both dead larvae which did not survive when emerging, as well as fully emerged alive adults, L:20-6-208.

172 Correction needed here – wasn’t the paper cited (line 174) about  Lestes sponsa, a damselfly with a different life history from I. elegans?  

>>>We have corrected that this research was performed on a different species, L:212-214. We also address the fact that in this experiment we did not verify  intrinsic mortality vs. mortality caused by cannibalism. We therefore pooled mortality caused by intrinsic and cannibalistic causes, L:28-216.

183 change to read ‘….damselflies whose legs and …’

>>>We have corrected this sentence, L: 225-226.

203 for clarity, change  ‘hatching phenology’ and temperature to read  ‘early versus late hatching, warming versus control and sex were fixed effects’…..This makes the dichotomy in each treatment clear.

>>>We have changed this sentence according to the suggestion, L:245-247.

203 remind the reader here what the response variables were – i.e. survivorship, age and mass at emergence, etc.

>>>We have added response variables to clarify the sentence, L:243-, :L249-250.

205 delete ‘the’ to read ‘….at emergence only…’

>>> We corrected this sentence, L:248.

206 add ‘models’ after ‘using linear mixed’.

>>>We added this, L:250.

208 change to read ‘…for multiple larvae sharing….’

>>>We corrected this sentence, L:252.

217-218 confusing sentence

>>>We changed this sentence for better clarity, L:260-262.

223 change ‘survival’ to ‘surviorship’ and end the sentence there; the rest is repetitive as cannibaslim was simply assumed.

>>>We changed this part of the sentence, L267-268. In other places, unless we do not discuss cannibalism, we changed cannibalism to survival/survivorship/survival rate, e.g. L: 22, L:97, L:108 and L:371.

223 replace hatching phenology with ‘Early versus late hatching ….

>>>We replaced this part of the sentence according to the suggestion, L:268-269.

228 delete ‘n’ to read ‘higher ratio’.

>>>We corrected this sentence, L:276.

228 Fig. 2B doesn’t show anything by sex

>>>We prepared additional figure in supplementary material and added reference to it in L:277.

Fig 2. In B can you show which contrasts were significant ? (maybe with an asterix)

>>>We have included data about contrasts in the text, L:271-275.

264 Please be more specific than ‘…life history and physiology’.

>>>We extended the first sentence of discussion by adding life history traits and immune function, L:317-319.

274 change to read ‘in contrast with …’

>>>We changed this sentence, L:329.

277-280 I would have thought adding a second generation to a season would result in lower mass and size at emergence because growth is speeded up. So at some point with global warming, even in Sweden it might no longer be beneficial.

>>>We added this potential scenario in L:344-346.

303 change to read ‘is rather rare in insects…’

>>>We changed this sentence according to the suggestion, L:363.

311  well, the difference between early and later hatching was only a 2-week difference. Perhaps tell us what % of the total growing season that is in Sweden, which would make it sound more significantly different.

>>> We have now used modelled temperatures to calculate the percentage of growth season that was covered in-between the two hatching dates (8%), see L:384-388.

Reviewer 2 Report

The authors tested how hatching date and increased temperature affected several life cycle and physiological traits of the cannibalistic damselfly Ischnura elegans. The authors reported that hatching date and temperature cause phenologial shifts did not affect cannibalism rate, rearing temperature affect voltinism, age and mass at emergence, but hatching date did not do so. Early hatchers had elevated phenoloxidase activity at emergence, as an indicator of investment of immune function.

This is a straightforward and interesting study. Below, I outline a few thoughts that, if addressed, would improve the manuscript. I hope my comments are helpful to the authors.

 1)  Introduction: The authors listed several prediction on the impact of hatching date or warm temperature on life history or physiology of this species, but little is explained as to why such a predction was achieved. It would be better to give basis for your prediction. For example, why the earlier hatcher is predicted to be bigger at emergence, despite lower age at emergence is also predicted?(L78-82, L270-271)  

2)Why you focused on immune function as a response variable?  Is there any general prediction about that?

3) L88-89: Please explain why warmer temperature increases cannibalism? The logic is not clear.

4) Please provide more detail on the overwintering stage (instar) of this species. Does the overwintering stage is fixed to a particular nymphal instar? Or all nymphal instar can pass the winter season? If latter is the case, time constraints on nymphal growth are relaxed.

5) Experimental design: in this experiment, the second winter was simulated by low temperature and short photoperiod of very short duration (1w). Is that enough for inducing and terminating second diapause? Inadequate induction or termination of second diapause would have some negative effect on the nymphal growth of the next season. I understand this procedure must shorten the duration of laboratory experiments, but need some explanation on this concern.

Author Response

The authors tested how hatching date and increased temperature affected several life cycle and physiological traits of the cannibalistic damselfly Ischnura elegans. The authors reported that hatching date and temperature cause phenologial shifts did not affect cannibalism rate, rearing temperature affect voltinism, age and mass at emergence, but hatching date did not do so. Early hatchers had elevated phenoloxidase activity at emergence, as an indicator of investment of immune function.

This is a straightforward and interesting study. Below, I outline a few thoughts that, if addressed, would improve the manuscript. I hope my comments are helpful to the authors.

  • Introduction: The authors listed several prediction on the impact of hatching date or warm temperature on life history or physiology of this species, but little is explained as to why such a predction was achieved. It would be better to give basis for your prediction. For example, why the earlier hatcher is predicted to be bigger at emergence, despite lower age at emergence is also predicted? (L78-82, L270-271)  

>>>We provided an additional reasoning for our predictions based on both hatching phenology, voltinism, and increased food competition L: 86-115.

2)Why you focused on immune function as a response variable?  Is there any general prediction about that?

>>>We focused on immune function as a response variable, because it is an important fitness-related physiological trait that is highly dependent on temperature and time constraints – two factors studied in this experiment. We added more information about phenoloxidase activity as an immune function in the introduction, see L:100-103, L:111-115.

3) L88-89: Please explain why warmer temperature increases cannibalism? The logic is not clear.

>>>We provided additional reasoning and logic for our prediction on decreased survivorship in general. Predicted decreased survival is mainly based on increased metabolism and activity of ectotherms under warming, L:108-1110. Note that in agreement with Reviewer 1, in many instances we changed the term cannibalism to survivorship or survival, with the later term being more relevant to our experimental methods.

4) Please provide more detail on the overwintering stage (instar) of this species. Does the overwintering stage is fixed to a particular nymphal instar? Or all nymphal instar can pass the winter season? If latter is the case, time constraints on nymphal growth are relaxed.

>>>This is a good comment. Based on  David J. Thompson data (1978) overwintering stage of this species happens primarily in instar nine and ten. However, overwintering in these instars occurs mainly in univoltine fraction of the population, indicating that semivoltine fraction overwinters in earlier instars, as shown by a no normal distribution of instars during winter period. Also, emergence is weakly synchronized in this species. The above suggest that at high latitudes time constraints are to some extent relaxed, unless larvae represent a fast cohort with a univoltine life cycle. We added this in the text, L:124-125.

5) Experimental design: in this experiment, the second winter was simulated by low temperature and short photoperiod of very short duration (1w). Is that enough for inducing and terminating second diapause? Inadequate induction or termination of second diapause would have some negative effect on the nymphal growth of the next season. I understand this procedure must shorten the duration of laboratory experiments, but need some explanation on this concern.

>>>A good point. We have added some explanation in the methods section. To check whether the diapause was initiated and terminated prior- and during the second winter, we have prepared a histogram of attempted emergences, see Supplementary Information, Fig. S1. The histogram shows a clear peak of emergence of semivoltine individuals about one month after a brief second winter had ended. This suggests that gradually decreasing photoperiod followed by a temperature decrease allowed larvae to start, develop and terminate winter diapause, and finally emerge shortly after. We have included this information in the description of the experimental design, L: 185-192.

Round 2

Reviewer 2 Report

I am satisfied that you have addressed the my comments adequately.